# The Effects of Process Parameters on the Porosity of a VBO Prepreg/Fiber-Interleaved Layup Composite

Yu-Wen Sheu and Wen-Bin Young *

Department of Aeronautics and Astronautics, National Cheng Kung University, Tainan 70101, Taiwan;
yovin.sheu@gmail.com
* Correspondence: youngwb@mail.ncku.edu.tw

**Abstract:** This research aimed to explore the porosity characteristics of a "hybrid" layup composite; this involved combining a fully impregnated prepreg and a dry fiber fabric via the the vacuum-bag-only (VBO) manufacturing process to create unidirectional carbon-fiber laminates sized at $15 \times 15$ cm$^2$. This investigation delved into several VBO process parameters encompassing the debulking technique, curing cycle, laminate saturation index, and thickness. The primary goal was to comprehend how these factors impacted the porosity levels within the laminate. Elevating the dwelling temperature during the curing cycle, employing a saturation index beyond 1.57, and utilizing thicker laminates emerged as strategies for decreasing the void content in the laminate. By implementing the optimal parameters identified through this research, we produced composite laminates that exhibited a substantial reduction in porosity. Furthermore, the study extended to modifying the two-stage curing cycle into a multi-stage cure cycle. This modification provided evidence that the incorporation of more dwell stages contributed to a further reduction in porosity. This study also featured a comparative analysis involving two types of laminates: one with prepreg fibers oriented at 0° and dry fibers oriented at 90° and another laminate with a sole 0° layup using a combination of prepreg and dry fibers. The findings suggest that the cross-layup allowed the prepreg fibers to conform more effectively to the protruding weft, thus eliminating voids induced by the weft. In conclusion, this research underscores the potential for a significant reduction in porosity within hybrid layup composites manufactured using the VBO process.

**Keywords:** vacuum bag only; carbon fiber composite; prepreg; porosity; hybrid layup

## 1. Introduction

In the modern aerospace domain, the creation of top-tier composite elements stands as a fundamental practice which is frequently accomplished through the utilization of autoclave prepregs. [1]. The traditional autoclave methodology involves the application of substantial consolidation pressure, typically within the range of 3–12 atm, to prepregs within an autoclave. This controlled pressure disrupts macro- and mesovoids due to trapped air or volatiles within the prepreg laminate. This fragmentation leads to their subdivision into smaller fragments that are then effectively extracted through resin flow. This intricate process yields components that are characterized by a formidable mechanical strength, minimal porosity, and exceptional uniformity [2,3]. However, the autoclave route carries inherent expenses linked to its acquisition and maintenance, coupled with the substantial power requirements and potential occupational hazards associated with high temperatures and pressures. These limitations underscore the need to explore alternatives to the autoclave process.

This motivation has driven the exploration of the Out-of-Autoclave (OOA) approach within the aerospace industry. Notable for its independence from the use of an autoclave during the fabrication process, the OOA approach offers many benefits, including cost-effectiveness, a reduced environmental impact, and heightened adaptability to diverse

component geometries. This avenue of research has been the subject of extensive investigations by scholars and researchers in recent decades. Among these alternative approaches, the vacuum-bag-only (VBO) process has emerged as a promising avenue. This process offers enhanced control over the volume content and orientation of fibers, making it an attractive, cost-effective alternative to traditional autoclaving. Unlike the autoclave process, the VBO process relies solely on a vacuum pressure of 1 atm to consolidate prepreg laminates. This significant departure from high consolidation pressures yields cost savings and allows for the adoption of economically viable curing methods such as heat ovens, blankets, and molds [4,5]. It is important to note that the lack of substantial consolidation pressure in the VBO process precludes the intricate expulsion mechanisms found in autoclave processing. A review of the out-of-autoclave process was reported in [6].

A pivotal component of the VBO process involves the use of specialized, semi-impregnated prepregs known as out-of-autoclave (OOA) prepregs. Unlike their fully impregnated counterparts, these prepregs feature engineered vacuum channels (EVaCs) integrated into the dry fiber regions. EVaCs enhance the in-plane air permeability of the laminate, thereby facilitating the escape of air, moisture, and volatiles during vacuum application [5,7]. Recent advancements have introduced diverse variations of OOA prepregs to enhance process stability and elevate the quality of the resulting components [4,8,9]. Empirical evidence underscores that under optimal processing conditions and when paired with OOA prepregs, the VBO process can yield composite components whose mechanical attributes are on par with those of their autoclave-manufactured counterparts [5,7]. The vacuum-bag technique can be used to manufacture primary structures such as decks, hulls, superstructures, and bulkheads and secondary structures such as partition panels and interior joint work [10].

However, the VBO process is sensitive to various process parameters, including debulking cycles, curing profiles, humidity levels, and air permeability characteristics [11–13]. The execution of the VBO process under suboptimal conditions can result in an elevated void content within components, consequently impacting crucial mechanical properties such as transverse tensile properties, bending characteristics, interlaminar shear strength, and elastic attributes; the risk of composite delamination is also accentuated under such circumstances [14–16]. A double-bagging process was also proposed to facilitate the degassing process from the collapse of prepreg stacking [17].

Numerous research studies have been conducted to identify optimal process parameters for the VBO technique, with the overarching goal of enhancing the robustness of the process and minimizing porosity within components. One approach involves the integration of heated debulking into the VBO process. While this enhancement augments out-of-plane air permeability, it concurrently introduces the risk of obstructing in-plane air evacuation [18]. A study conducted by Ridgard et al. [19] demonstrated this principle by subjecting laminate samples to a 50 °C heated debulk for 4 h before curing. The resulting components exhibited porosity levels comparable to those achieved through a 16 h room-temperature debulk. Hu et al. [18] delved deeper into the underlying mechanisms, establishing a correlation between out-of-plane permeability, debulk temperature, and prepreg fiber structure. Elevated debulk temperatures were associated with a lower resin viscosity, thereby facilitating the expulsion of inter-laminar voids through the out-of-plane direction. However, this simultaneous infiltration of resin into dry fiber regions obstructed in-plane air evacuation. Their conclusions underscored the applicability of heated debulking to woven fiber laminates with length-to-thickness ratios exceeding 5.5, while unidirectional fiber laminates with lower thicknesses (<1 mm) benefited the most. Maguire and colleagues delved into the significance of prepreg formats and the production process for VBO prepregs [20]. The manual application of epoxy powder was examined for its potential to result in an uneven distribution of powder, potentially resulting in improved laminate uniformity. The research findings validated the theory that epoxy powder serves as a preventive measure against exothermic reactions in thick composite materials. Edward and his colleagues developed a unidirectional semi-prepreg aimed at enhancing

the reliability of VBO processing [21]. They utilized a fortified epoxy resin for this purpose. The semi-prepreg was tailored to halt the distribution of resin. Consequently, there was an enhancement in through-thickness permeability, enabling more effective gas removal. Laminates created using the semi-prepreg exhibited fewer imperfections compared to those formed using traditional VBO prepregs. The morphology of the resin's characteristics was noted as a crucial factor influencing the formation of defects.

In a study by Mujahid et al. [22], a manufacturer-recommended two-stage cure cycle (MRCC) was employed as a benchmark. Recognizing the influence of residual stress and strain effects, two modified cure cycles were introduced. The EMRCC inserted an additional curing stage into the original MRCC progression. At the same time, the DC incorporated an elevated first-stage dwelling temperature. The DC yielded a notable 18.12% reduction in through-thickness voids compared to the MRCC. Furthermore, the EMRCC and DC resulted in tensile strengths 20% and 17.4% higher than the MRCC, respectively.

Hyun et al. [23] leveraged resin cure kinetics and viscosity modeling, using the "effective flow number" concept to optimize the reference cure cycle. Their modifications, such as the replacement of the 16 h room-temperature debulk with a 2h 60 °C heated debulk (Modified I) and the substitution of the 121 °C dwelling stage with a prolonged 177 °C dwelling stage (Modified II), resulted in the components produced via the Modified II cycle displaying the highest degrees of cure and tensile strength.

It is imperative to highlight that utilizing OOA prepregs within the VBO process entails high costs, extended lead times, and minimum order quantity requirements. Addressing these challenges, Yang et al. [24] devised a hybrid layup laminate for the VBO process, incorporating fully impregnated prepregs alongside dry fiber fabrics. The inclusion of dry fabrics mimics the action of partially impregnated VBO prepregs, facilitating air and volatile evacuation. Chang et al. [25] manufactured unidirectional carbon fiber laminates measuring $50 \times 50$ cm$^2$ using the above hybrid laminate which demonstrated tensile strengths akin to their autoclave-manufactured counterparts and minimized thickness deviations. However, it should be noted that the flexural strength of the hybrid laminate was 16% lower due to its heightened porosity, which resulted from protruding dry fabric wefts. This research initiative aims to meticulously analyze the intricate interplay between multiple VBO process parameters and laminate porosity within a prepreg/dry fiber laminate. The goal is to enhance the quality of components for their prospective applications.

## 2. Methodology

Experiments were carried out in three phases to comprehend the impacts of various process parameters on the hybrid layup laminate parts. In each step, unidirectional carbon fiber panels measuring $15 \times 15$ cm$^2$ were produced using the VBO process, employing different process parameters for quality assessment.

### 2.1. Prepreg/Fiber Hybrid Layup

A unidirectional carbon fiber prepreg UD150 (37 wt% resin) and dry fabrics provided by Wah Hong Industrial Corp, Taiwan, were used in this research. The laminate arrangement design principle was established by Chang et al. [25]. The carbon fiber prepreg and dry fabrics were layered alternatively, with the dry fabrics acting as the EVaCs between the prepregs. During the consolidation process using vacuum pressure, it is essential to ensure that the excess resin in the prepregs is enough to fill the dry fabrics. In the design process, the laminate's saturation index ($S_{index}$) is a critical factor to consider. The $S_{index}$ reflects the ratio between the resin and the porosity within the laminate structure. Previous research has shown that achieving an oversaturated state ($S_{index} > 1$) is essential to ensuring the effective filling of laminate dry fiber regions with resin. The expression for the $S_{index}$ is provided below:

$$S_{index} = \frac{V_r}{V_{pore,atm}} \tag{1}$$

where $V_r$ represents the total volume of resin in the laminate, and $V_{pore, atm}$ represents the total volume of voids under 1 atm of pressure on the laminate. Based on the required $S_{index}$ and laminate thickness, the number of layers of prepregs and dry fabrics, along with the stacking sequence, can be determined.

### 2.2. VBO Process of Producing a Hybrid Layup Laminate

The unidirectional carbon fiber laminates used for the quality assessment were fabricated using the VBO process, as illustrated in Figure 1. Initially, the prepreg was left at room temperature for 30 min to de-ice it, preventing the accumulation of moisture in the laminate and reducing the viscosity of the resin for stacking. The prepreg was then cut into a rectangular shape measuring $15 \times 15$ cm$^2$, while the dry fiber was cut into $15 \times 16$ cm$^2$; the additional 1 cm of dry fiber would contact the breather layer to create a continuous air channel, enhancing interlaminar air evacuation.

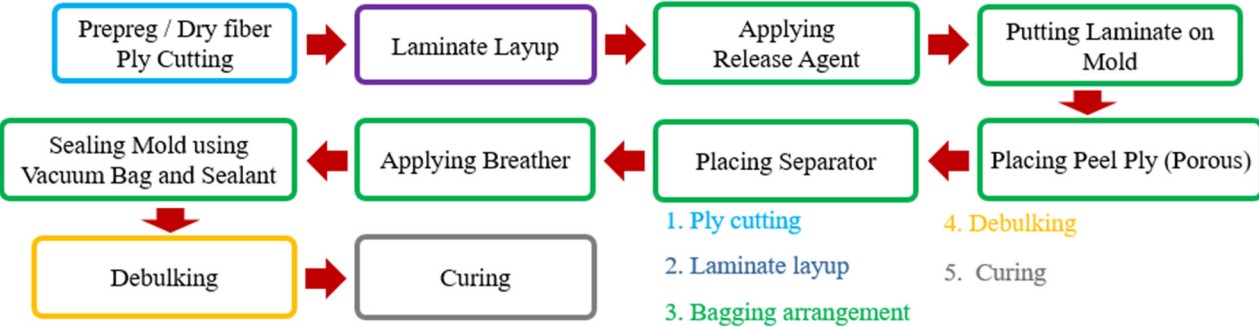

**Figure 1.** VBO manufacturing process.

The next step entailed stacking the prepreg and dry fiber in the designated sequence to form a hybrid layup laminate. A plastic blade was used to scrape the laminate surface, preventing the entrapment of air bubbles during the layup. Simultaneously, a release agent was applied to the mold and left to dry completely for 40 min. Subsequently, the laminate was placed on the mold, followed by the peel ply, separator, and breather. The entire mold assembly was then sealed using a vacuum bag and sealant.

Once sealed, the assembly was subjected to vacuum pressure to debulk, removing the air, moisture, and volatiles inside the laminate. Finally, the laminate was heated based on the selected cure cycle, ensuring that the resin thoroughly impregnated the dry fiber area and ultimately reached a fully cured state.

### 2.3. VBO Process Parameters

For the VBO process parameters, the cure cycle, saturation index, debulking method, and laminate thickness were chosen as the parameters to be studied, given their easily adjustable natures and potential to significantly influence part quality. The standard parameters listed in Table 1 were derived from previous research and served as a baseline for subsequent parameter variations where P is one layer of prepreg, Pb is two layers of prepreg, and F is the dry fabric.

**Table 1.** Standard parameters served as a baseline.

| | |
|---|---|
| **Cure Cycle** | **120 °C $\times$ 2 h + 160 °C $\times$ 2 h** |
| Saturation Index | 1.27 |
| Debulking method | Room Temperature (25 °C) for 1 h |
| Laminate Arrangement | Pb/F/Pb/F/Pb |

1 All prepregs and dry fiber fabrics oriented at 0°; 2 P is one layer of prepreg, and Pb is two layers of prepreg; 3 F is the dry fabric.

First, a comparison was conducted between two-stage cure cycles employing different dwelling temperatures, as listed in Table 2. The elevated dwelling temperature (140 °C) yielded a lower minimum resin viscosity but a shorter gel time. This translates into an improved resin flow yet a limited time for infiltration into the dry fiber area. Conversely, the reduced dwelling temperature (100 °C) had the opposite effect. Throughout this comparison, all other parameters remained constant.

**Table 2.** Cure cycles comparison.

| Cure Cycle | First Stage | Second Stage |
|---|---|---|
| Decreased Dwelling Temperature | 100 °C × 2 h | 160 °C × 2 h |
| Standard | 120 °C × 2 h | 160 °C × 2 h |
| Raised Dwelling Temperature | 140 °C × 2 h | 160 °C × 2 h |

Secondly, various saturation indexes ($S_{index}$) were compared. Previous research established that the laminate needs to be oversaturated ($S_{index} > 1$) to fill all voids adequately. Different $S_{index}$ values, including 1.27 (standard), 1.57, 1.69, and 1.92, were selected to further investigate the impact of excessive resin on part quality. As shown in Table 3, The varying $S_{index}$ values were achieved by adjusting the number of prepregs in the laminate arrangement. Again, all other parameters were kept consistent during this comparison.

**Table 3.** Saturation indexes with different layups.

| Saturation Index | Laminate Arrangement |
|---|---|
| 1.27 (Standard) | Pb/F/Pb/F/Pb |
| 1.57 | Pc/F/Pb/F/Pc |
| 1.69 | Pc/F/Pc/F/Pc |
| 1.92 | Pb/Pb/F/Pc/F/Pb/Pb |

1 P is one layer of prepreg, and Pb is two layers of prepreg; 2 F is the dry fabric.

Lastly, three debulking methods—a standard debulk, an elongated debulk, and a heated debulk—were compared to assess the effectiveness of longer debulk times or higher debulk temperatures, as listed in Table 4. Different laminate arrangements, including standard, elevated $S_{index}$, and double thickness, were also included in the comparison, as shown in Table 5. All other parameters remained constant throughout this comparison.

**Table 4.** Comparison of debulking methods.

| Debulking Method | Temperature | Time |
|---|---|---|
| Standard | Room Temperature (25 °C) | 1 h |
| Elongated | Room Temperature (25 °C) | 7 h |
| Heated | 50 °C | 1 h |

**Table 5.** Different laminate arrangements.

| Saturation Index | Laminate Arrangement |
|---|---|
| 1.27 (Standard) | Pb/F/Pb/F/Pb |
| 1.27 (Double Thickness) | [Pb/F/Pb/F/Pb]$_s$ |
| 1.57 | Pc/F/Pb/F/Pc |

1 P is one layer of prepreg, and Pb is two layers of prepreg; 2 F is the dry fabric.

### 2.4. Multi-Stage Curing Cycle

Table 6 shows that the optimized parameters concluded from the previous section were adopted as the standard parameters, forming a baseline for modifications to the cure cycle. The initial step of the cure cycle was increased to 140 C, and the laminate arrangement with saturation index of 1.57 was used. To establish a three-stage cure cycle, a lower temperature stage (100 °C) was introduced before the initial step in the two-stage cure cycle. A four-stage cure cycle was also developed by introducing a new location at 120 °C to ensure a smoother temperature progression. Table 7 shows all the multi-stage cure cycles. The aim of incorporating of an additional cure cycle stages was to provide the resin with an extended gel time, allowing more time for the infiltration of resin into the dry fiber area.

**Table 6.** Standard parameters for the two-stage cure cycle.

| Cure Cycle | 140 °C × 2 h + 160 °C × 2 h |
|---|---|
| Saturation Index | 1.57 |
| Debulking Method | Room Temperature (25 °C) for 1 h |
| Laminate Arrangement | $[Pc/F/Pb/F/Pc]_S$ |

**Table 7.** Multi-stage cure cycles.

| Two-stage cure cycle | First Stage | | | Second Stage |
|---|---|---|---|---|
| | 140 °C × 2 h | | | 160 °C × 2 h |
| Three-stage cure cycle | First Stage | Second Stage | | Third Stage |
| | 100 °C × 1 h | 140 °C × 1 h | | 160 °C × 2 h |
| Four-stage cure cycle | First Stage | Second Stage | Third Stage | Fourth Stage |
| | 100 °C × 0.5 h | 120 °C × 0.5 h | 140 °C × 1 h | 160 °C × 2 h |

### 2.5. Fiber Layup Orientation

The optimized parameters and the most favorable cure cycles from previous sections were established as the standard parameters, as shown in Table 8. This phase explored the influence of different fiber orientations between the prepreg and dry fiber fabrics on the quality of the laminate. Laminates featuring prepreg fibers at 0° and dry fibers at 90° were compared with laminates utilizing a sole 0° layup, as shown in Table 9.

**Table 8.** Standard parameters with multi-stage cure cycle.

| Cure Cycle | 100 °C × 0.5 h +120 °C × 0.5 h + 140 °C × 1 h + 160 °C × 2 h |
|---|---|
| Saturation index | 1.57 |
| Debulking method | Room Temperature (25 °C) for 1 h |
| Laminate arrangement | $[Pc_{0°}/F_{0°}/Pb_{0°}/F_{0°}/Pc_{0°}]_S$ |

**Table 9.** Layup with different fiber orientation.

| Fiber Orientation | Fiber Arrangement |
|---|---|
| 0° layup | $[Pc_{0°}/F_{0°}/Pb_{0°}/F_{0°}/Pc_{0°}]_S$ |
| Cross layup (0° prepreg/90° dry fiber) | $[Pc_{0°}/F_{90°}/Pb_{0°}/F_{90°}/Pc_{0°}]_S$ |

### 2.6. Quality Assessment Methods

Three assessment methods—microscopy, an average thickness assessment, and a laminate resin loss assessment—were utilized to comprehend the varying levels of part

quality. From each produced laminate, four specimens were cut and polished, as shown in Figure 2 for the left, right, up, and down parts.

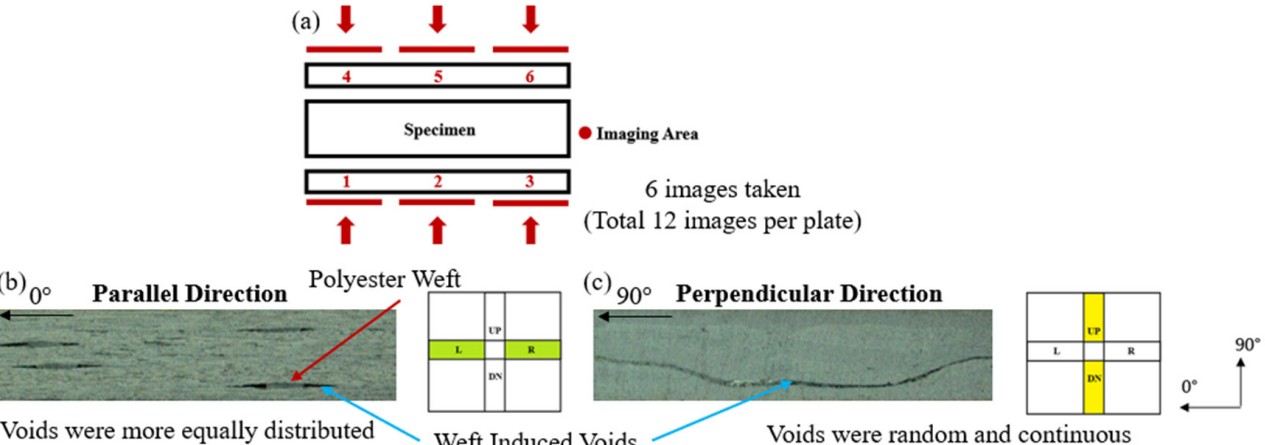

**Figure 2.** Digital microscopy; (**a**) specimen imaging locations; (**b**) void seen from the direction parallel to the fibers; (**c**) voids seen from the direction perpendicular to the fibers; images were enlarged by ×22.5.

Digital microscopy with a magnification of 22.5 was employed to capture images of the specimen cross-sections, as shown in Figure 2a. These images were then calibrated using OpenCV and processed through imageJ. In the processing, the images were converted to grayscale to identify voids. The porosity in each photograph represents the void area's percentage of the total image area. The average porosity of a laminate was calculated from the average porosity in the specimen images while excluding outliers. For accuracy, only specimens with cross-sections parallel to the fibers (L and R samples from the laminate) were used for porosity calculations due to the void shape characteristics of hybrid layup laminates, as shown in Figure 2b,c. Since most voids were located near the dry fiber region, the images were centered in this area.

To determine laminate thickness, specimens were measured at the same 10 points (40 points in total for L, R, UP, and DN samples) to obtain a precise average laminate thickness, as shown in Figure 3. For laminates using an oversaturated laminate ($S_{index} > 1$), the estimated thickness $h_c$ without considering resin loss was the sum of the laminate resin, prepreg fiber, and dry fiber lumped thicknesses. The expression for the estimated laminate thickness $h_c$ is provided below:

$$h_c = \tilde{h}_{df} + \tilde{h}_{pf} + \tilde{h}_r \tag{2}$$

where $\tilde{h}_r$ is the resin lumped thickness, $\tilde{h}_{pf}$ is the prepreg fiber lumped thickness, and $\tilde{h}_{df}$ is the dry fiber lumped thickness. The estimated fiber volume content $v_{f\_est}$ can be calculated using the lumped thicknesses of the laminate fibers and the estimated laminate thickness $h_c$, and is expressed as below:

$$v_{f\_est} = \frac{\tilde{h}_{df} + \tilde{h}_{pf}}{h_c} \tag{3}$$

By measuring the average laminate thickness $h_{c,actual}$, we can then calculate the actual fiber volume content, as shown below:

$$v_{f\_actual} = \frac{\tilde{h}_{df} + \tilde{h}_{pf}}{h_{c,actual}} \tag{4}$$

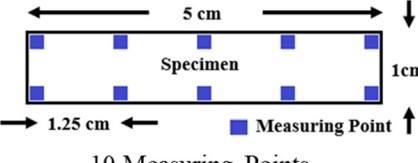

**Figure 3.** Locations for measuring specimen thickness.

To compare the laminate resin loss under different process conditions, the weights of the laminates before and after curing were measured. The laminate resin loss ratio $w_{resin\ loss}$ was defined as below:

$$w_{resin\ loss} = \frac{W_{initial} - W_{final}}{W_{initial}} \times 100\% \tag{5}$$

where $w_{initial}$ is the weight of the laminate before curing, and $w_{final}$ is the laminate's weight after curing.

## 3. Result and Discussion

### 3.1. Effect of VBO Process Parameters

When comparing different first-stage dwelling temperatures in the cure cycle, the results of the experiments, as depicted in Figure 4, revealed a decreasing trend in porosity as the dwelling temperature was raised. Utilizing a higher dwelling temperature of 140 °C showed a significant reduction of 0.4% in void content compared to the standard (120 °C). Conversely, using a lower dwelling temperature of 100, °C increased the porosity by 0.36% compared to the standard temperature. Upon observing the specimen cutaway images in Figure 5, it was noted that the voids predominantly existed in the areas surrounding dry fiber wefts. Additionally, the void dimensions decreased as the dwelling temperature was elevated to 140 °C. This outcome demonstrates that in a prepreg/dry fiber hybrid layup laminate, decreasing the resin viscosity by raising the dwelling temperature can enhance the infiltration of resin into the fiber tows, ultimately reducing the porosity.

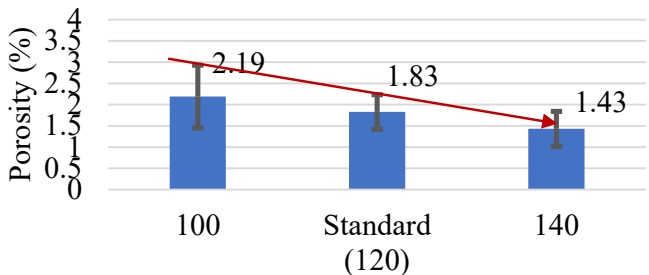

**Figure 4.** Porosity comparison for different cure cycles.

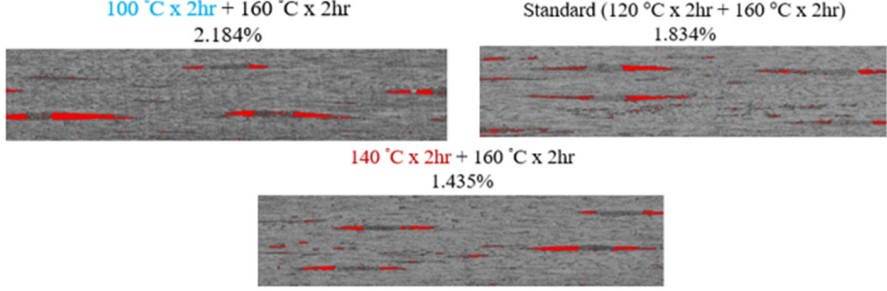

**Figure 5.** Comparison of microscopic images of cure cycles (red areas indicate voids; images were enlarged by ×22.5).

The quality of laminates with different saturation indexes was subsequently compared. The laminate with $S_{index}$ = 1.57 exhibited a substantial 1.21% reduction in porosity compared to the laminate with $S_{index}$ = 1.27, as shown in Figure 6. As the saturation index increased to 1.92, porosity began to rise. However, it remained 0.88% lower than the standard $S_{index}$ = 1.27. Upon observing the specimen cutaway images in Figure 7, it becomes evident that the primary change occurred in the quantity and size of the voids surrounding the dry fiber wefts. The polymer weft induced significant thickness deviations in localized areas, resulting in fiberless holes that demand ample resin for complete filling.

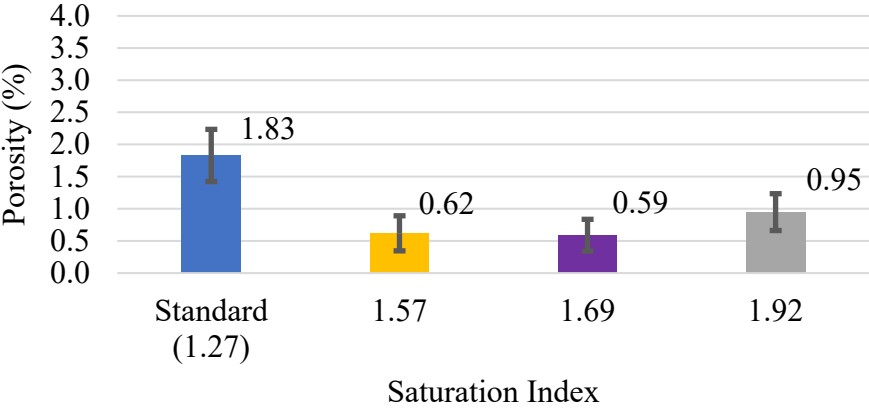

**Figure 6.** Porosity of laminates with different saturation indexes.

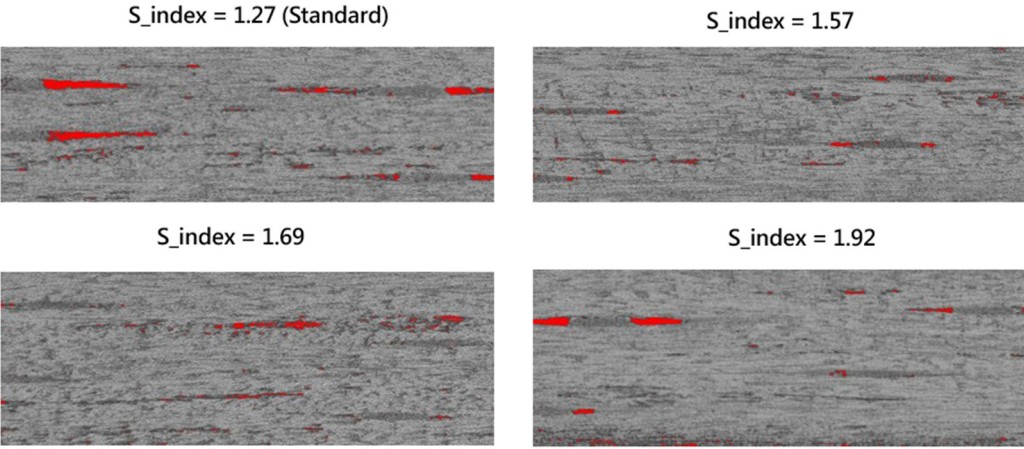

**Figure 7.** Microscopic images of laminates with different saturation indexes (red areas indicating voids; images were enlarged by ×22.5).

The average thickness was first measured to calculate the fiber volume contents of laminates with different $S_{index}$ values; during manufacturing, the laminate resin was under free-bleed conditions. As shown in Figure 8, the measured fiber volume content did not show significant change with the increased saturation index. On the other hand, the estimated fiber volume content of the laminate decreases with an increase in the saturation index (a reduction of about 7.4% for a saturation index of 1.93). Notice that the calculation of the estimated fiber volume content was based on the assumption of a condition under which no resin bled out. With the higher saturation index, there is more resin inside the laminate, which results in less laminate compaction [25]. Therefore, the thickness of the laminate will increase with the saturation index, resulting in a lower estimated fiber volume content based on Equation (3). During the fabrication process, some resin bled out along the edges. As the saturation index increased, more resin bled out from the laminate under compaction until the fibers were fully compressed by the vacuum pressure, resulting in

fewer changes in the laminate thickness. Thus, no large variations in the measured fiber volume content are found.

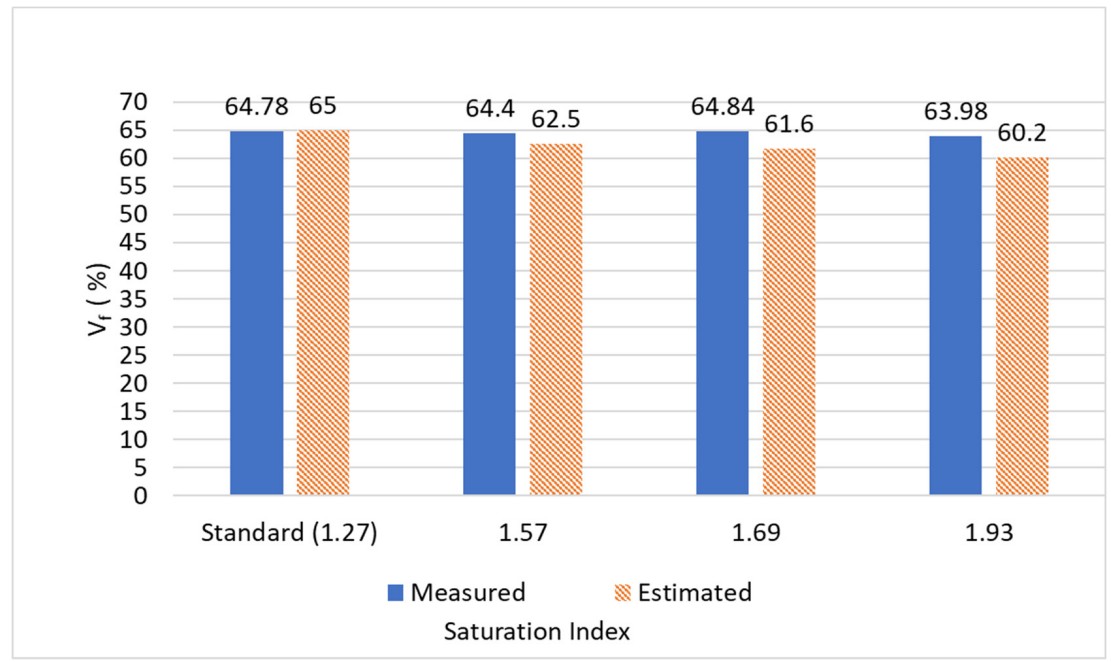

**Figure 8.** Fiber volume contents of laminates with different saturation indexes.

As shown in Figure 9, when comparing various debulking methods, it was observed that elongated debulk times failed to reduce porosity for the standard fiber arrangement. This observation indicates that the expected debulk duration effectively eliminated air, vapor, and volatiles within the laminate. Conversely, the heated debulking method yielded a 0.79% higher porosity than the standard method. This increase is likely due to the premature infiltration of resin into the EVaCs, obstructing the evacuation of air, water vapor, and volatiles. In the case of the raised $S_{index}$ fiber arrangement, the situation was comparable, with elongated debulk times failing to yield significant changes in porosity. The heated debulking method, in this case, led to a 0.3% increase in porosity compared to the standard way.

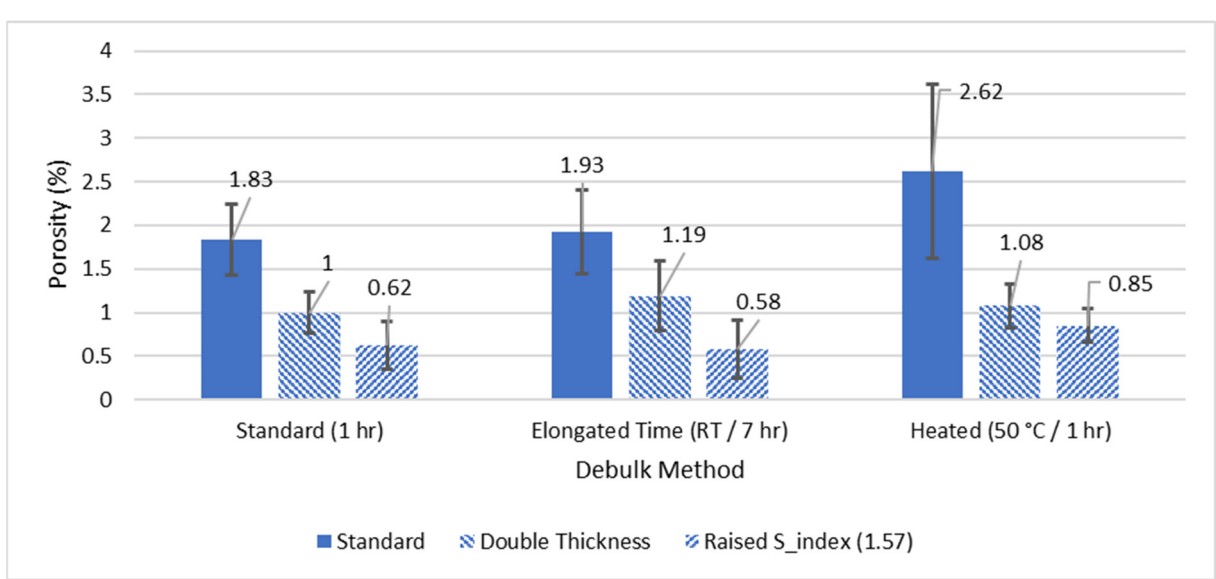

**Figure 9.** Laminate porosity values achieved via different debulking methods.

Different debulking methods exhibited minimal effects on porosity regarding the double-thickness fiber arrangement. This suggests that thicker laminates (2.6 mm) display greater resilience to changes in debulking conditions. Parts with a double-thickness arrangement showed lower porosity levels (1~1.19%) compared to the standard format (ranging between 1.83 and 2.62%). As shown in Figure 10, the double-thickness arrangement also achieved a slightly higher fiber volume content (66.16~66.73%) than the estimated fiber volume content and the standard format (64.78~66.11%). This outcome could be attributed to a reduction in the void content, enabling more efficient resin filling.

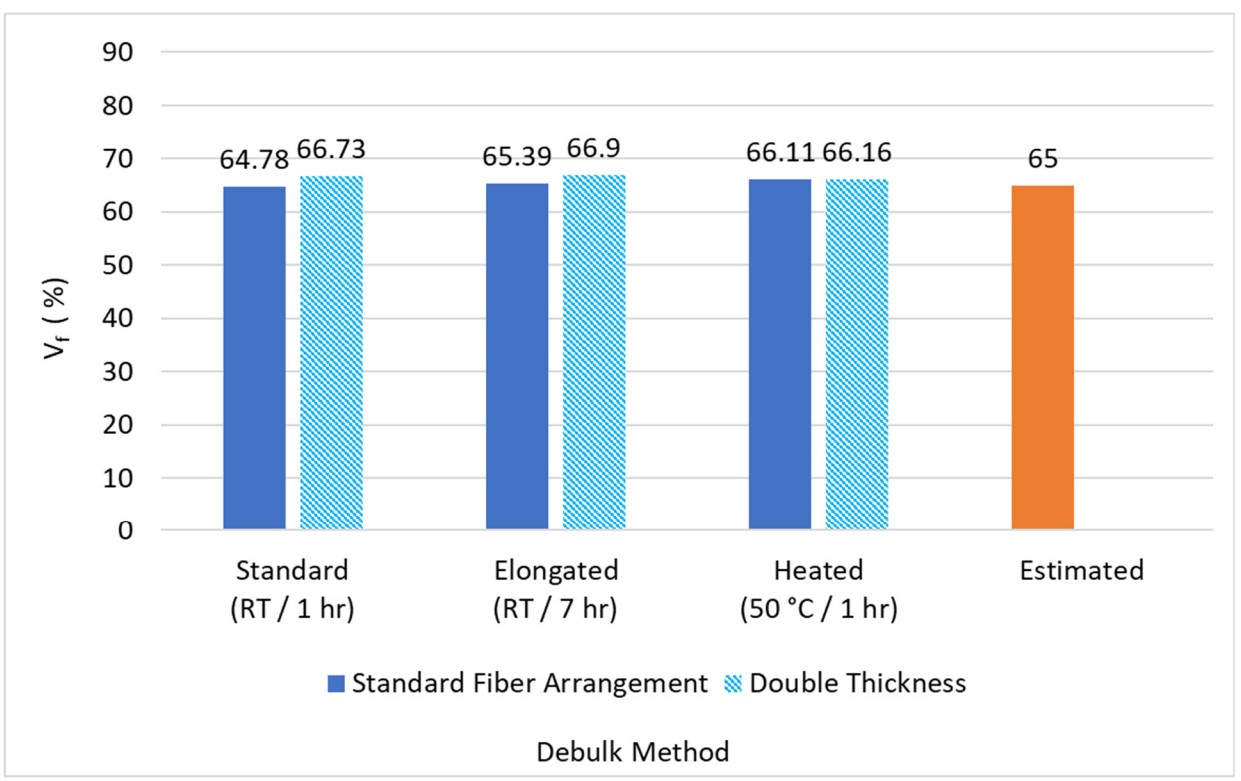

**Figure 10.** Fiber volume contents of laminates produced via different debulking methods.

*3.2. Effect of Multi-Stage Curing*

The optimized process parameters were employed as the standard for the multi-stage cure cycle. As illustrated in Figure 11, the laminate produced using the standard parameters exhibited a porosity of 0.542%. This outcome underscores the effectiveness of utilizing the optimized process parameters in which a cure cycle with a higher initial step temperature and a laminate arrangement with a saturation index of 1.57 were used. Introducing a three-stage cure cycle further improved the void content to 0.45%. Subsequently, a four-stage cure cycle reduced the void content even further to just 0.23%, with only a minimal presence of weft-induced voids, as depicted in Figure 12. These results affirm that adopting a multi-stage cure cycle to extend the resin flow time (gel time) provides the resin in the prepregs with ample opportunity to infiltrate the dry fiber area effectively and fill more voids. Moreover, the increased number of cure stages generates a smoother temperature curve, mitigating the risk of overheating the resin.

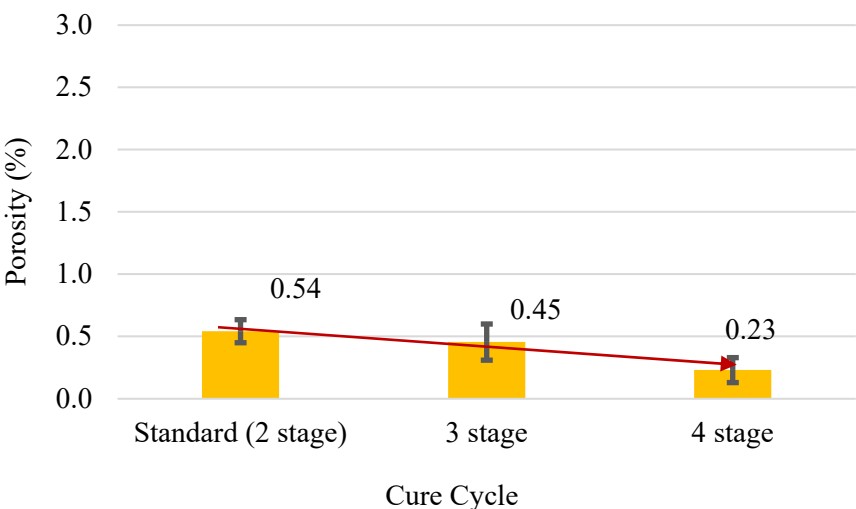

**Figure 11.** Comparison of porosity values for multi-stage cure cycles.

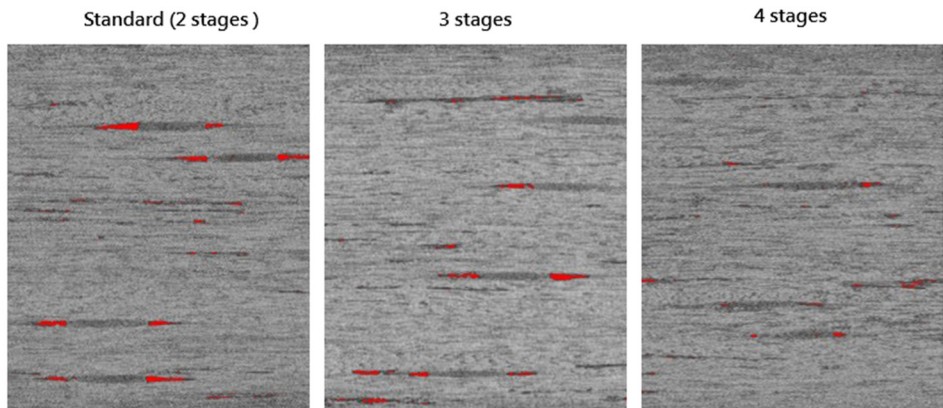

**Figure 12.** Comparison of multi-stage cure cycle microscopic images (red areas indicating voids; images were enlarged by ×22.5).

Compared to the standard two-stage cure cycle, the multi-stage (three/four) cure cycle produced laminates which exhibited a slightly reduced thickness, as depicted in Figure 13. This outcome is likely attributable to the improved infiltration of resin into the dry fiber tow and voids. Consequently, the fiber volume content showed a slight increase, as shown in Figure 14. Furthermore, the laminate weight loss ratio decreased with the addition of more cure cycle stages, as illustrated in Figure 15. This reduction indicates an enhanced infiltration of resin into the dry fiber tow and voids, resulting in less resin loss.

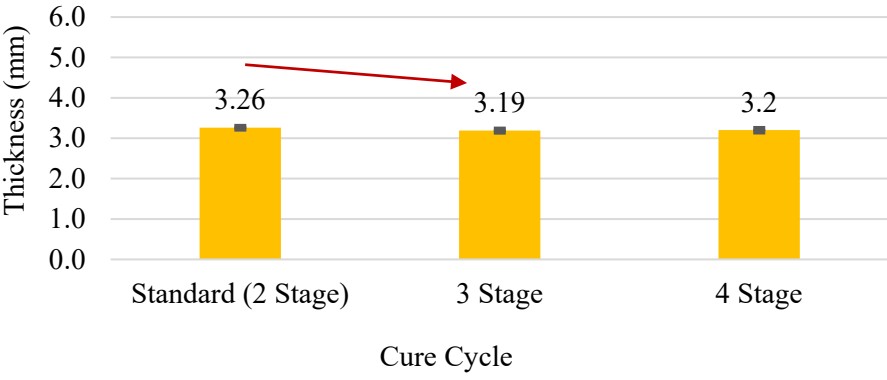

**Figure 13.** Comparison of laminate thickness values for multi-stage cure cycles.

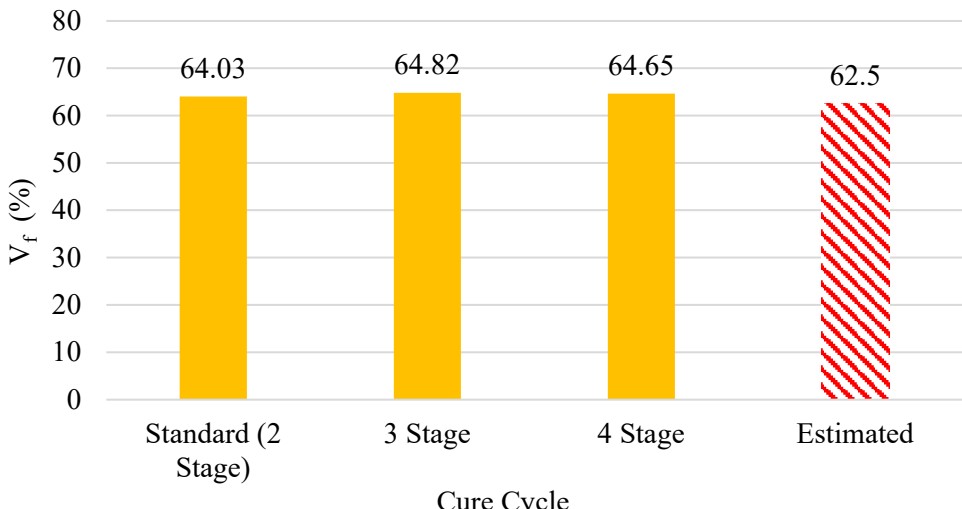

**Figure 14.** Comparison of fiber volume content values for multi-stage cure cycles.

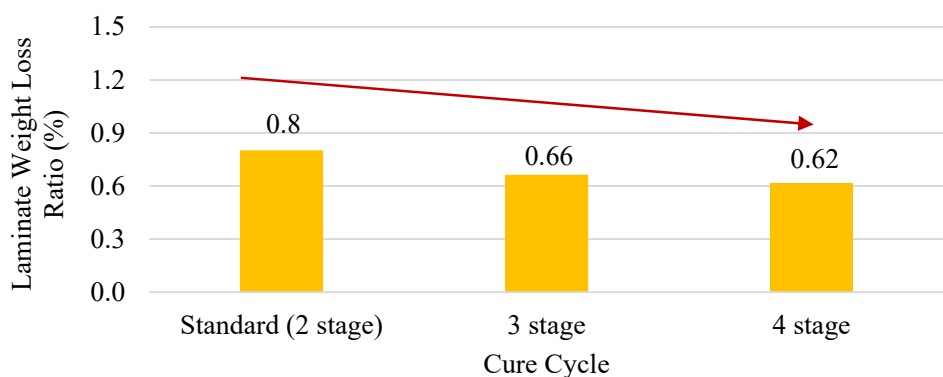

**Figure 15.** Comparison of laminate weight loss ratios for multi-stage cure cycles.

### 3.3. Effect of Laminate Fiber Orientation

The optimized process parameters listed in Table 8 were used as the standard conditions for comparing various laminate fiber orientations. As depicted in Figure 16, unlike all 0° fiber arrangements, a configuration of 0° prepreg/90° dry fiber significantly reduced the void content from 0.23% to a mere 0.14%. This reduction can be attributed to the alignment of the prepreg fibers at 0° and the weft polyester of dry fibers at 90°, effectively conforming the prepreg fibers to the protruding weft. This alignment eliminated weft-induced voids, as demonstrated in Figure 17. These holes formed around the dry fibers may not be completely filled with resin during the consolidation stage, which results in voids in the composites.

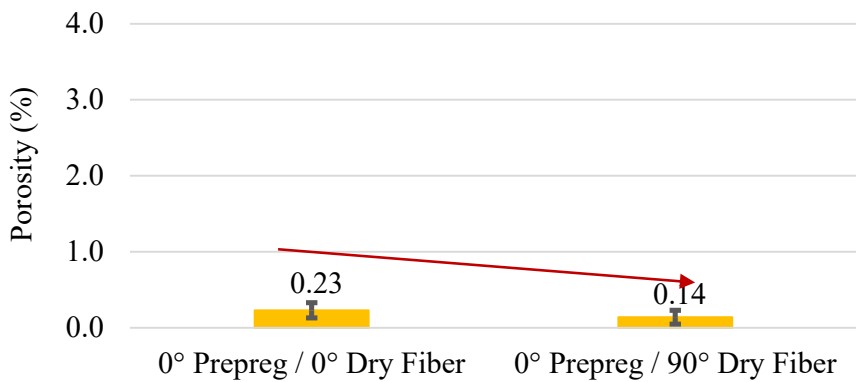

**Figure 16.** Comparison of porosity values of laminates containing fibers with different orientations.

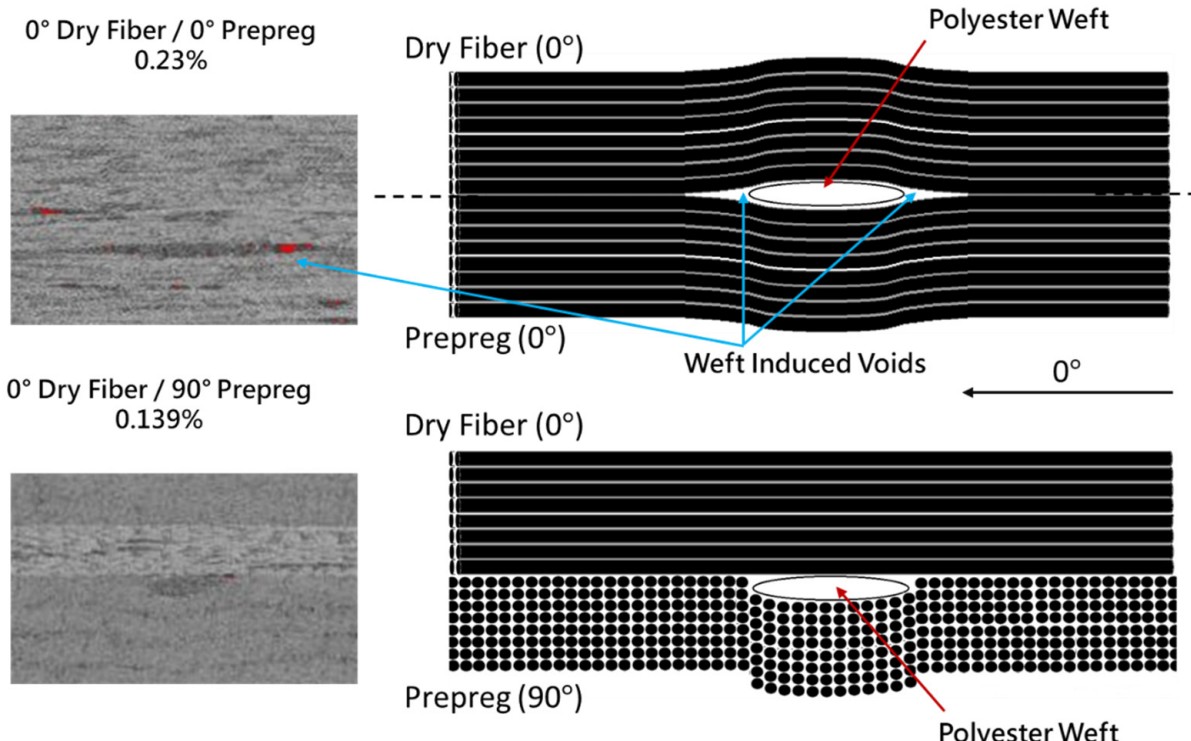

**Figure 17.** Comparison of microscopic images of different fiber orientations (red areas indicating voids; images were enlarged by ×22.5).

As illustrated in Figures 18 and 19, the configuration with a 0° prepreg/90° dry fiber arrangement exhibited a slightly increased thickness compared to all the 0° fiber arrangements. This variation can be attributed to different fiber orientations within the laminate. Consequently, the fiber volume content showed a slight decrease as a result. The laminate weight loss ratio was also elevated for the 0° prepreg/90° dry fiber arrangement, as demonstrated in Figure 20. This increase might be attributed to a higher amount of resin bleed-out occurring along the prepreg fiber direction into the breather rather than effectively infiltrating the dry fiber/breather connection area.

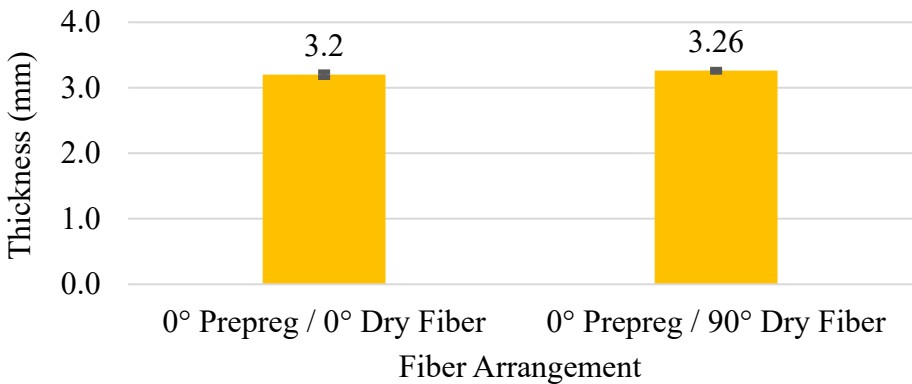

**Figure 18.** Comparison of laminate thickness values for different fiber orientations.

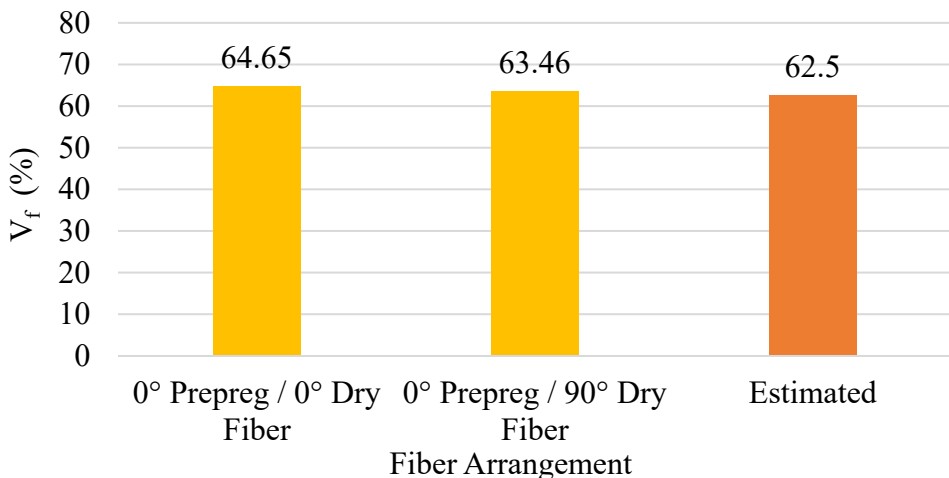

**Figure 19.** Comparison of fiber volume content for different fiber orientations.

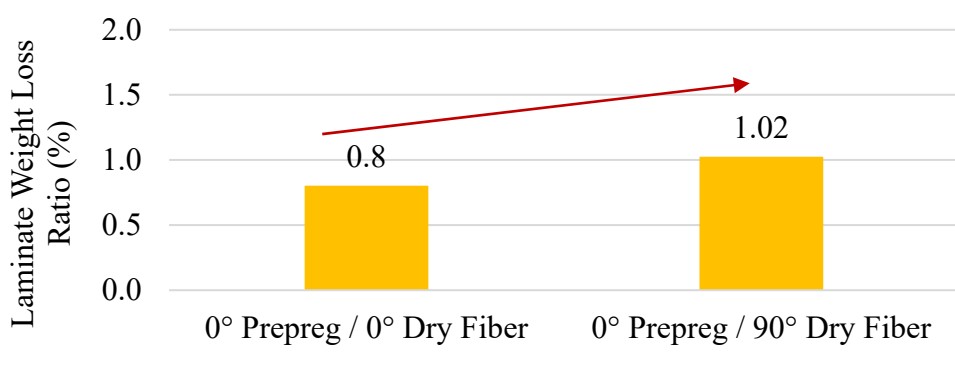

**Figure 20.** Comparison of laminate weight loss ratios for different fiber orientations.

## 4. Conclusions

This research substantiates the impact of VBO manufacturing parameters and cure cycles on the quality of unidirectional carbon prepreg/dry fiber hybrid laminates, successfully reducing porosity and yielding high-quality laminates. In the initial stage, elevating dwelling temperatures (140 °C) during the cure cycle decreased resin viscosity, thus enhancing the infiltration of resin into fiber tows and reducing the laminate's porosity. $S_{index}$ values above 1.57 indicate diminished porosity due to increased resin filling in regions with high thickness deviations and fiberless sections. Heated debulking is unnecessary, resulting in premature resin infiltration, blocking EVaCs, and resulting in higher degrees of porosity in thinner laminates (1.6 mm and 2.6 mm). Increasing the number of cure cycle stages extends the duration of the resin's low-viscosity state, enhancing the impregnation of dry fiber tows and reducing porosity. Aligning the prepreg fibers with the polymer weft on the dry fiber fabric eliminates weft-induced voids, showcasing the feasibility of tailoring prepreg/dry fiber orientations in laminate production.

In conclusion, to produce unidirectional carbon fiber composite laminates, it is recommended to maintain an $S_{index}$ of at least 1.57 to ensure sufficient resin content in the laminate. Employing a higher initial dwelling temperature (140 °C) for the cure cycle, followed by a four-stage cure cycle modification, ensures optimal resin infiltration. If achieving minimal porosity is a priority, stacking prepreg and dry fabric with a 90° directional difference is advisable.

**Author Contributions:** Conceptualization, writing—review and editing, supervision, W.-B.Y.; methodology, validation, formal analysis, data curation, writing—original draft preparation, Y.-W.S. All authors have read and agreed to the published version of the manuscript.

**Funding:** This research received funding from Ministry of Science and Technology in Taiwan under the contract of number MOST 108-2221-E-006-068.

**Institutional Review Board Statement:** Not applicable.

**Informed Consent Statement:** Not applicable.

**Data Availability Statement:** Not applicable.

**Conflicts of Interest:** The author declares no conflict of interest.

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
