# Peer review of "The Effects of Process Parameters on the Porosity of a VBO Prepreg/Fiber-Interleaved Layup Composite"

_jcs, doi:10.3390/jcs7100412_

Round 1

Reviewer 1 Report

The subject of this research is the analysis of the manufacturing process parameter on porosity VBO prepreg composite material. The investigation delved into several VBO process parameters, encompassing debulking technique, curing cycle, laminate saturation index, and thickness. The subject of the research is described quite extensively, moreover, supported by relevant graphics, it forms a whole, thoroughly explaining the investigated models. The obtained conclusions are presented in a logical and coherent manner for the reader. However, some parts of the proposed manuscript need improvement:

- the introduction is quite short and has only a few current literature items, I recommend adding current literature references

Author Response

Several recent literatures were added in the context which was hilighted.

Reviewer 2 Report

In this work, the authors studies the effect factors in reducing porosity in the hybrid layup composite manufactured using the vacuum bag only process. In general, this work provides several useful data and it is recommended to publish after a minor revision. Here are some suggestions and comments:

1. Kindly include a reference for the statement concerning S_index >1 found in lines 127-130 on page 3.

2. It is advisable for the authors to include a scale or ruler in all microstructure images.

3. Consider changing the pattern fill used in the charts to ensure their clarity when printed in black and white.

4. When presenting changes in the data, it would be beneficial for the authors to provide the quantity values by calculating the different percentages.

5. Figure 17 effectively depicts the appearance of voids, so the authors should provide a discussion or explanation regarding the formation of these voids.

Author Response

  1. A reference was added for the S_index > 1.
  2. Add the magnification in figures. Images were x22.5 enlarged
  3. Add patterns for the charts with multiple columns.
  4. We do some modifications. However, for the quantity change in percentage, we will not address the change in percentage to avoid confusion with the percentage quantity itself, such as volume fraction, porosity, and weight loss ratio.
  5. These formed holes around the dry fibers may not be completely filled with resin during the consolidation stage, resulting in composites voids.

Round 2

Reviewer 1 Report

All comments carried as I suggested

Reviewer 2 Report

The reviewer doesn't have any comments over the revised version, and recommends to publish this work. Congratulations!